# Exploring the Readiness of Critical Care in Implementing Continuous Renal Replacement Therapy in Hail Hospitals, Saudi Arabia: Findings for Acute Kidney Injury Patient Care Improvement

**DOI:** 10.3390/healthcare13182320

**Published:** 2025-09-16

**Authors:** Sameer A. Alkubati, Abdulaziz A. Alfaraaj, Mokhtar A. Almoliky, Salman H. Alsaqri, Khalil A. Saleh, Bahia Galal Siam

**Affiliations:** 1Department of Medical Surgical Nursing, College of Nursing, University of Hail, Hail 55476, Saudi Arabia; aalfaraaj@moh.gov.sa (A.A.A.); m.amed@uoh.edu.sa (M.A.A.); s.alsaqri@uoh.edu.sa (S.H.A.); k.saleh@uoh.edu.sa (K.A.S.); b.siam@uoh.edu.sa (B.G.S.); 2Intensive Care Unit, King Salman Hospital, Hail Health Cluster, Hail 55471, Saudi Arabia

**Keywords:** continuous renal replacement therapy, CCNs, knowledge, attitude, practice, critical care, Saudi Arabia

## Abstract

**Background/Objectives:** Continuous renal replacement therapy (CRRT) is a critical intervention for managing acute kidney injury (AKI) in critically ill patients. Critical care nurses (CCNs) play a pivotal role in its implementation, requiring adequate knowledge, a positive attitude, and practice proficiency. This study aimed to assess the levels and factors affecting CCNs’ knowledge, attitudes, and practices (KAPs) regarding the care of patients receiving CRRT in Hail Hospitals, Saudi Arabia. **Methods:** A cross-sectional study was conducted with 190 registered CCNs from the critical care units of Hail Hospitals, Saudi Arabia, from March to May 2025. Data were collected using a sociodemographic characteristics sheet and the knowledge, attitudes, and practices questionnaire on CRRT. Non-parametric statistical tests (Mann–Whitney U, Kruskal–Wallis, and Spearman’s correlation tests) were used to determine the relationships between variables. A multiple linear regression analysis was used to explore the factors affecting the study variables. **Results:** The majority of CCNs had a high level of knowledge regarding CRRT management (66.3%), followed by moderate (21.1%) and low (16.6%) levels. Additionally, the majority had a high level of attitude regarding CRRT management (74.7%), followed by moderate (18.4%) and low (6.9%) levels. The majority of CCNs had a high level of practice regarding CRRT management (66.8%), followed by low (23.7%) and moderate (9.5%) levels. There was a significant positive correlation between CCNs’ knowledge, attitudes (rs = 0.230, *p* < 0.001), and practices (rs = 0.192, *p* < 0.001). Additionally, there was a significant positive correlation between CCNs’ attitudes and practices (r = 0.419, *p* < 0.001). Multiple linear regression revealed a significant model (*p*  <  0.001) when sociodemographic and work-related factors were analyzed as predictors of CCNs’ levels of knowledge, attitudes, and practices regarding CRRT: Having more experience, working with a nurse-to-patient ratio of 1:2, and frequently using CVVHDF as a CRRT modality were significant factors for higher knowledge levels when compared to the reference categories. In addition, having a bachelor’s degree and frequently using CVVHD, CVVHDF, or SCUF as a CRRT modality were significant factors for higher attitude levels regarding CRRT when compared to the reference categories. Having a bachelor’s degree and frequently using CVVHD or CVVHDF as a CRRT modality were significant factors for higher practice levels regarding CRRT when compared to the reference categories. **Conclusions:** CCNs in Hail Hospitals self-reported high levels of knowledge, attitudes, and practices regarding CRRT management, followed by moderate levels. Targeted educational programs, standardized protocols, and organizational support are recommended to improve CCN care for CRRT and to optimize patient outcomes.

## 1. Introduction

Continuous renal replacement therapy (CRRT) is a vital treatment approach used in critical care settings to treat patients with hemodynamic instability, severe chronic renal disease, and acute kidney injury (AKI) [1,2,3]. International guidelines, including the KDIGO Clinical Practice Guideline for Acute Kidney Injury, recommend CRRT as the preferred modality for hemodynamically unstable ICU patients with AKI, reflecting its global recognition as a lifesaving therapy [2]. In contrast to intermittent hemodialysis, which is usually administered in short, intensive sessions, CRRT is administered continuously over 24 h, enabling gradual fluid and solute removal to maintain metabolic and fluid balance in critically ill patients [4]. Because it reduces circulatory stress and promotes organ recovery, this ongoing strategy is especially beneficial for patients with severe fluid overload, uremic toxicity, or acid–base abnormalities [5]. CRRT incorporates various modalities, including continuous venovenous hemofiltration (CVVH), continuous venovenous hemodialysis (CVVHD), continuous venovenous hemodiafiltration (CVVHDF), and slow continuous ultrafiltration (SCUF), each of which aims to meet specific clinical needs [6]. As up to 50% of critically ill patients may develop AKI and require renal replacement treatment in intensive care units (ICUs), the use of CRRT has expanded globally [7]. In Saudi Arabia, the aging population, rising rates of chronic illnesses (such as diabetes and hypertension), and increasing burden of critical care admissions are some of the factors contributing to the comparably large prevalence of AKI [8].

CCNs play an essential role in the effective implementation of CRRT, as they are in charge of a variety of duties, such as CRRT machine setup and calibration, patient monitoring, treatment parameter adjustments, alarm troubleshooting, and the handling of obstacles such as tubing obstructions or vascular access problems [9,10]. The complexities of these duties necessitate not only technical expertise but also a thorough understanding of CRRT concepts, a favorable attitude toward their application, and adherence to evidence-based procedures [11]. Effective CRRT delivery has been shown to improve patient survival rates and reduce complications; however, its success hinges on the competence and preparedness of nursing staff [12].

Although CCNs play a vital role, research indicates that many struggle to learn CRRT. Studies show that CCNs often do not know enough about CRRT practices, such as alarm management or anticoagulation techniques, which can cause treatment delays or poor patient outcomes [10,13,14]. To be precise, research consistently indicates that CCNs’ knowledge of CRRT is often inadequate in complex areas, such as alarm management, anticoagulation therapy, and complication management [10,13]. Yu et al. (2024) [10] surveyed 405 ICU staff members across 66 hospitals in China, reporting a mean knowledge score of 51.46 ± 5.96, suggesting a moderate understanding but significant variability. Factors such as higher education, CRRT experience, and specialist panel membership were associated with better knowledge, underscoring the importance of targeted training [10]. Similarly, Alsolami and Alobaidi (2024) found that hemodialysis CCNs in Saudi Arabia had a low mean knowledge score (6.4 ± 2.0 out of 12, or 53.3%) regarding vascular access management, with only 37% having received specialized training [15].

Furthermore, attitudes toward CRRT implementation may vary, influenced by factors such as perceived complexity, prior training, and organizational support [16]. Negative attitudes or a lack of confidence can hinder CCNs’ engagement with CRRT, thus affecting care quality [10]. Additionally, inconsistencies in clinical practice, such as variations in protocol adherence or machine operation, have been reported, often owing to differences in training, experience, or institutional policies [12,17,18]. CCNs’ attitudes toward CRRT are critical, as they influence their engagement and willingness to adopt complex therapies. Yu et al. (2024) reported positive attitudes among Chinese ICU staff (mean score: 58.71 ± 2.19), driven by factors such as experience and training [10]. Several organizational and individual factors influence CCNs’ ability to implement CRRT effectively. Organizational factors such as high nurse-to-patient ratios, staffing shortages, and limited access to CRRT machines or training programs hinder performance [19]. In addition, inconsistent institutional policies, a lack of specialist panels, and standardized guidelines further complicate CRRT delivery [20].

Individual factors, such as CCNs’ education, experience, and confidence, significantly impact knowledge and practice. Alsolami and Alobaidi (2024) found that CCNs with bachelor’s degrees were 92% more likely to be knowledgeable about vascular access (*p* < 0.05) [15]. Similarly, Sagiron and Jarelnape (2022) reported a significant correlation between educational level and knowledge of hemodialysis procedures (*p* = 0.01) [17]. In Saudi Arabia, the demand for CRRT is rising in parallel with the increasing prevalence of AKI, largely driven by chronic conditions such as diabetes and hypertension. CCNs play a central role in delivering CRRT, including in machine setup, patient monitoring, alarm troubleshooting, and complication management. This study is the first to systematically assess CCNs’ readiness for CRRT implementation in Saudi Arabia, where the burden of AKI is rapidly increasing. By identifying strengths in knowledge and attitudes, alongside gaps in practice and organizational support, the findings provide an evidence base for developing targeted training, staffing policies, and modality-specific protocols. Highlighting these areas is crucial not only for improving local patient outcomes but also for informing global efforts to optimize CRRT delivery in critical care settings. Therefore, this study addressed the following research questions: What are the levels of knowledge, attitudes, and practices of CCNs toward CRRT? What factors influence these domains? We hypothesized that a higher educational level, longer professional experience, and greater exposure to CRRT modalities would be positively associated with readiness. Accordingly, the objectives of this study were to assess CCNs’ knowledge, attitudes, and practices regarding CRRT in Saudi Arabia and to identify factors that influence these outcomes.

## 2. Methods

### 2.1. Study Design

A cross-sectional design was used to assess CCNs’ knowledge, attitudes, and practices regarding CRRT implementation in Hail Hospitals, Saudi Arabia, from March to May 2025. This design enabled a snapshot of CCNs’ competencies at a single point in time, which was suitable for descriptive and correlational analyses.

### 2.2. Setting and Sample

This study was conducted across critical care departments, including the intensive care unit (ICU), critical care unit (CCU), emergency room (ER), and artificial kidney unit (AKU), at King Salman Specialist Hospital and King Khalid Hospital in Hail, Saudi Arabia. The King Salman Specialist Hospital’s critical care departments comprise 65 CCNs in the ER, 35 in the ICUs, 30 in the CCUs, and 15 in the AKU. Additionally, King Khalid Hospital’s ER, ICU, CCU, and AKU departments employ 65, 35, 33, and 60 CCNs, respectively, and they are involved in the management of CRRT cases. Each hospital consists of seven Prismaflex CRRT machines, which are used to manage critically ill patients requiring renal replacement therapy. Although CRRT is typically initiated and managed in the ICU and CCU, nurses working in the ER were included in this study. In the participating hospitals, ER nurses frequently manage critically ill patients who are potential candidates for CRRT, and they are responsible for initial patient stabilization, preparation, and handover to ICU staff. Their inclusion was therefore justified to provide a comprehensive assessment of readiness among all nurses engaged in the continuum of care for patients requiring CRRT.

A sample size of 181 CCNs was determined using the OpenEpi web-based calculator, Version 3.01 (www.openepi.com), based on the following criteria: a 95% confidence level, 5% precision, and a population size of 338 CCNs familiar with CRRT. However, 190 registered CCNs were included in the study. The inclusion criterion was being a registered nurse currently working in a critical care unit with prior experience with CRRT. CCNs on official leave, on vacation, or participating in training programs during the data collection period were excluded from the study. This convenience sampling approach was chosen to ensure the inclusion of CCNs with relevant experience while accounting for staff availability during the survey period. All CCNs were invited to participate; of them, 20 were on vacation, and 32 refused to participate. However, the questionnaire was distributed to 286 CCNs, of whom 190 completed the questionnaire, achieving a response rate of 66.4% (Figure 1).

### 2.3. Data Collection Tools

Data were collected using demographic and professional sheets and the validated knowledge, attitudes, and practices questionnaire on CRRT developed by Yu et al., 2024 [10]. The demographic and professional sheets included data such as age, sex, nationality, experience, qualifications, department, job position, patient-to-nurse ratio, shift hours, and extra work. In addition, CRRT practice patterns, such as the frequency of tubing blockage, CRRT mode, treatment modality, and experience level, were included. The knowledge, attitudes, and practices questionnaire on CRRT is divided into three parts. The first part focuses on CCNs’ knowledge of CRRT, and it includes 14 items. This section evaluates CCNs’ knowledge of basic CRRT concepts, including machine settings, anticoagulant strategies, vascular access preparation, therapy commencement, and complication management. Each item includes a 5-point Likert scale, with “1” meaning highly unfamiliar and “5” meaning very familiar. The total score ranges from 14 to 70 points. Higher self-rated scores correlate with higher knowledge levels. The second part investigates CCNs’ attitudes toward CRRT, using 15 items on a 5-point Likert scale, with “1” meaning “strongly disagree” and “5” meaning “strongly agree.” Higher total scores, which range from 15 to 75, indicate a more positive outlook [10].

The third part addresses CCNs’ practices regarding CRRT, with “yes” answers worth two points and “no” answers worth one point. Higher scores indicate higher levels of self-rated practice, and the total score varies from 10 to 20. This questionnaire was tested and validated in a previous study by 10 clinical experts who evaluated its content validity, and the total content validity was found to be 0.900 [10]. Additionally, Cronbach’s α coefficients for knowledge, attitudes, and practices were 0.956, 0.831, and 0.751, respectively, thus confirming the reliability of the questionnaire [10]. We conducted a pilot study with 20 CCNs to assess its clarity, comprehension, and relevance. Feedback from the pilot confirmed that all items were clearly understood and contextually appropriate. No terminological or conceptual difficulties were reported. The reliability of the knowledge, attitude, and practice domains was confirmed by Cronbach’s α, which was 0.822, 0.789, and 0.970, respectively. To facilitate interpretation, the total KAP scores were categorized into three levels: low, moderate, and high. This approach followed the cut-off strategy applied in the validated CRRT KAP questionnaire by Yu et al. (2024) [10] and related KAP studies. In our dataset, these thresholds also approximated percentile-based divisions, where “low” corresponded to scores below the 25th percentile, “moderate” corresponded to those in the 25th–75th percentile, and “high” corresponded to those above the 75th percentile [10].

The CCNs were given surveys and informed consent forms by the researchers during their breaks. All respondents received a thorough explanation of the study’s goals and methods prior to their involvement. Before accessing the survey, the participants had to read and sign an informed consent form. To protect confidentiality and response independence, all participants were informed that their participation was voluntary and anonymous. Department heads facilitated distribution but had no access to the responses. The surveys were returned directly to the researchers. The participants were assured that neither their supervisors nor the hospital administrators would be able to link their responses to their identity. Additionally, they were made aware of their freedom to leave the study at any time, without facing any repercussions.

### 2.4. Ethical Consideration

Ethical approval was obtained from the University of Hail’s Institutional Review Board (IRB) (No. H-2025-633). The participants provided informed consent, and confidentiality was ensured through anonymized data collection. The CCNs were informed of their right to withdraw without any consequences. The data were securely stored and accessible to the research team.

### 2.5. Data Analysis

The data were analyzed using IBM SPSS Statistics version 27.0. Continuous variables (knowledge, attitude, and practice scores) are summarized as medians and interquartile ranges due to their non-normal distributions (confirmed by the Shapiro–Wilk test, *p* ≤ 0.05). Categorical variables are described as frequencies and percentages. Non-parametric tests (Mann–Whitney U and Kruskal–Wallis) were used to assess differences in scores across sociodemographic and professional groups. Spearman’s correlation was used to examine the relationships among knowledge, attitudes, and practices. To explore factors of the KAP scores, multiple linear regression models were initially applied. Prior to analysis, diagnostic checks were conducted, including assessments of multicollinearity (variance inflation factor, VIF < 2), the normality of residuals, and model fit (adjusted R^2^, ANOVA F-test). In addition, a sensitivity analysis using ordinal logistic regression was conducted to account for the ordinal nature of the total KAP scores. The ordinal regression results were consistent with the linear regression results, confirming the robustness of the findings. Standardized β coefficients, 95% confidence intervals, and *p*-values are reported for regression models. Statistical significance was set at *p* < 0.05.

## 3. Results

### 3.1. Sociodemographic of the Participating CCNs

Table 1 summarizes the sociodemographic characteristics of the participating CCNs. A total of 190 CCNs were included, with slightly more than half aged 32 years or younger (52.1%) and 47.9% aged older than 32 years. The majority of the participants were male (69.5%) and Saudi nationals (75.8%). In terms of professional experience, 56.3% of the CCNs had seven years of experience or less, while 43.7% had more than seven years of experience. Regarding educational qualifications, 81.6% held a bachelor’s degree, 8.9% held a diploma, and 9.5% held a master’s degree. Concerning workplace departments, 35.3% of the CCNs worked in the intensive care unit (ICU), 22.6% worked in the emergency department (ER), 20.0% worked in the critical care unit (CCU), and 22.1% worked in the artificial kidney unit (AKU). In terms of job positions, the majority were staff CCNs (67.9%), followed by charge CCNs (26.3%), and a smaller proportion were head CCNs or supervisors (5.8%). The patient-to-nurse ratios varied, with the most common being 1:2 (37.9%), followed by 1:4 (26.8%), 1:3 (20.5%), and 1:1 (14.7%).

Regarding shift duration, 61.1% of the CCNs worked 12-h shifts, while 38.9% worked 8-h shifts. The majority (87.9%) reported not engaging in extra work outside of their main job, whereas 12.1% reported having additional work responsibilities.

### 3.2. CRRT Practice Patterns and CRRT Mode Used Among CCNs

Table 2 outlines the CRRT practice patterns and experiences of the participating CCNs. Regarding the frequency of tubing blockage during CRRT, 44.2% of the CCNs reported that they had never observed tubing blockage, 40.0% indicated that it sometimes occurred, 11.1% reported that it frequently occurred, and 4.7% stated that no attention was given. Concerning the CRRT mode most frequently used, the majority of the CCNs (65.3%) reported using continuous venovenous hemodiafiltration (CVVHDF), followed by continuous venovenous hemodialysis (CVVHD) at 14.2%, continuous venovenous hemodiafiltration (CVVH) at 10.5%, and slow continuous ultrafiltration (SCUF) at 10.0%.

### 3.3. Levels of CCNs’ Knowledge, Attitude, and Practice Scores Toward CRRT

As described in Figure 2, the majority of the CCNs had a high level of knowledge regarding CRRT management (66.3%), followed by moderate (21.1%) and low (16.6%) levels. Additionally, the majority had a high level of attitude regarding CRRT management (74.7%), followed by moderate (18.4%) and low (6.9%) levels. The majority of the CCNs had a high level of practice regarding CRRT management (66.8%), followed by low (23.7%) and moderate (9.5%) levels.

As shown in Table 3, there was a significant positive correlation between the CCNs’ knowledge and attitudes (rs = 0.230, *p* < 0.001) and practices (rs = 0.192, *p* < 0.001). There was a significant positive correlation between the CCNs’ attitudes and practices (r = 0.419, *p* < 0.001).

### 3.4. CCNs’ Knowledge, Attitude, and Practice Based on Sociodemographic and Work-Related Characteristics

Table 4 presents the differences in the mean ranks of the CCNs’ knowledge, attitude, and practice scores based on various sociodemographic and professional characteristics. CCNs older than 32 years had significantly higher practice scores (median = 10, IQR = 5–10) than those aged ≤32 years (median = 8, IQR = 4–9; *p* < 0.001). CCNs with >7 years of experience also showed higher knowledge (median = 49, IQR = 48–49 vs. 47, IQR = 37–49; *p* < 0.001) and practice scores (median = 10, IQR = 6–10 vs. 8, IQR = 8–10; *p* < 0.001). Similarly, 12-h shift CCNs had higher knowledge (median = 49, IQR = 48–50 vs. 46, IQR = 38–49; *p* < 0.001) and practice scores (median = 10, IQR = 5–10 vs. 8, IQR = 4–10; *p* = 0.035).

Educational qualification was strongly associated with all domains. CCNs with a bachelor’s degree had higher knowledge (median = 48, IQR = 47–49), attitude (median = 58, IQR = 57–59), and practice scores (median = 10, IQR = 7–10) than those with a diploma (knowledge = 37, IQR = 24–59; attitude = 45, IQR = 29–73; practice = 0, IQR = 0–0) and master’s degree (knowledge = 39, IQR = 13–47; attitude = 51, IQR = 26–63; practice = 5, IQR = 0–7; all *p* < 0.05). Departmental differences were significant for knowledge (*p* < 0.001) and attitude (*p* = 0.029), with ICU (median = 49, IQR = 46–51) and CCU CCNs (median = 49, IQR = 48–51) scoring higher than AKU CCNs (median = 48, IQR = 34–48).

Practice scores differed by job position, with staff CCNs (median = 10, IQR = 7–10) scoring higher than charge CCNs (median = 6, IQR = 0–8) and supervisors (median = 6, IQR = 3–9; *p* < 0.001). The patient-to-nurse ratio was associated with significant differences in knowledge (*p* = 0.001) and practice scores (*p* = 0.041), with the highest practice scores observed in 1:1 assignments (median = 10, IQR = 7–10).

The CRRT modality most frequently used showed highly significant differences across all domains (all *p* < 0.001). CCNs using CVVHDF recorded the highest knowledge (median = 49, IQR = 48–49), attitude (median = 58, IQR = 58–59), and practice scores (median = 10, IQR = 7–10), while those using CVVH and SCUF recorded the lowest scores.

### 3.5. Factors Affecting Knowledge, Attitude, and Practice Toward CRRT Among CCNs

Multiple linear regression revealed a significant model (*p*  <  0.001) when sociodemographic and work-related factors were analyzed as predictors of CCNs’ knowledge, attitudes, and practices toward CRRT. Variances of 29.7%, 23.7%, and 34.1% in CCNs’ knowledge, attitudes, and practices, respectively, were explained by the model (R^2^ = 0.345, adjusted R^2^ = 0.297; R^2^ = 0.270, adjusted R^2^ = 0.237; and R^2^ = 0.390, adjusted R^2^ = 0.341, respectively). For knowledge, significant positive predictors were having more than seven years of experience (β = 4.76, 95% CI [0.84, 8.69], *p* = 0.018), working with a patient-to-nurse ratio of 1:2 (β = 4.13, 95% CI [0.33, 7.93], *p* = 0.033), and most frequently using the CVVHDF (β = 11.70, 95% CI [6.33, 17.07], *p* < 0.001) or SCUF mode (β = 7.33, 95% CI [0.39, 14.26], *p* = 0.038) compared to using CVVH as the reference.

For attitude, significant positive predictors were holding a bachelor’s degree compared to a diploma (β = 7.92, 95% CI [1.60, 14.24], *p* = 0.014) and most frequently using the CVVHD (β = 7.60, 95% CI [1.71, 13.49], *p* = 0.012), CVVHDF (β = 12.24, 95% CI [7.08, 17.41], *p* < 0.001), or SCUF (β = 12.19, 95% CI [5.18, 19.20], *p* = 0.001) mode compared to the CVVH reference category.

For practices, significant positive predictors were holding a bachelor’s degree (β = 4.64, 95% CI [2.43, 6.85], *p* < 0.001) and most frequently using the CVVHD (β = 2.00, 95% CI [0.10, 3.91], *p* = 0.039) or CVVHDF (β = 2.70, 95% CI [0.84, 4.57], *p* = 0.005) mode compared to the CVVH reference category (Table 5).

## 4. Discussion

The findings of this study provide critical insights into the knowledge, attitudes, practices, and needs of CCNs in Hail Hospitals, Saudi Arabia, regarding the implementation of continuous renal replacement therapy (CRRT). The results reveal high levels of knowledge and attitudes, followed by moderate levels, but they highlight significant gaps in practice proficiency, aligning with the global and regional literature on CRRT implementation challenges [10,20].

### 4.1. Knowledge of CRRT

This study revealed that CCNs’ levels of knowledge and understanding of CRRT principles were high, followed by moderate, which is consistent with studies reporting variable knowledge levels in critical care settings. For instance, Yu et al. (2024) found a mean knowledge score of 51.46 ± 5.96 among ICU staff in China, suggesting that, while CCNs grasp basic CRRT concepts, advanced competencies—such as alarm management, anticoagulation adjustments, or complication management—may be lacking [10]. In Iraq, most CCNs (80.43%) demonstrated moderate CRRT knowledge, with only 4.35% exhibiting good knowledge [21]. This is particularly concerning given the complexity of CRRT, which requires precise knowledge to prevent treatment interruptions or patient harm [22]. The significant differences in knowledge scores by experience (*p* < 0.001), with CCNs with over seven years of experience scoring higher, align with the findings of Sagiron and Jarelnape (2022), who observed that education and experience enhanced hemodialysis knowledge (*p* = 0.01) [17].

CCNs with bachelor’s degrees also demonstrated higher knowledge scores (mean rank: 101.73, *p* = 0.001) than those with diplomas or master’s degrees, partially supporting Hypothesis 1. This finding echoes that of Alsolami and Alobaidi (2024), who reported that undergraduate-educated CCNs were 92% more likely to be knowledgeable about vascular access (*p* < 0.05) [15]. However, the lower knowledge scores among master’s degree holders (mean rank: 55.22) were unexpected and may reflect limited CRRT-specific training at the postgraduate level or a focus on administrative rather than clinical roles. The higher knowledge scores among ICU and CCU CCNs and CVVHDF users suggest that exposure to specialized settings and commonly used modalities enhances familiarity, which is consistent with the findings of Yu et al. (2024), who noted better KAP scores in teaching hospitals with structured CRRT programs [10].

Moderate and low levels of knowledge indicate the need for targeted education, particularly in areas such as alarm handling and complication management, where gaps are common [13]. In Saudi Arabia, where CRRT use is increasing due to the rising prevalence of AKI [8], these gaps could compromise patient outcomes, underscoring the urgency of comprehensive training programs.

### 4.2. Attitudes Toward CRRT

The high and moderate levels of attitude toward CRRT generally reflect positive attitudes toward CRRT implementation, aligning with the findings of Yu et al. (2024), who reported a mean attitude score of 58.71 ± 2.19 among Chinese ICU staff [10]. The moderate positive correlation between knowledge and attitude supports the assertion of the KAP model that knowledge shapes attitudes [23,24]. In a recent study conducted by Kadhim et al. (2025), a strong positive correlation was found between knowledge and attitude (r = 0.696, *p* < 0.0001) among CCNs working with CRRT [21]. CCNs with a greater understanding are likely to feel more confident and be receptive to CRRT, as seen in a study conducted by Al Qahtani and Almetrek (2017), where positive attitudes toward infection control (85.59 ± 8.09) were linked to better knowledge [25].

The significant differences in attitude scores by qualification (*p* = 0.035), with bachelor’s degree holders scoring higher, suggest that undergraduate education fosters positive perceptions, possibly because of its clinical focus. Similarly, CCU CCNs exhibited the highest attitude scores, likely because of their frequent exposure to CRRT in high-acuity settings. CVVHDF users also showed more positive attitudes, reflecting familiarity with a commonly used modality (65.3% of CCNs reported using CVVHDF). These findings indicate that experience and context shape attitudes, and they are consistent with those of another study that found that attitudes toward infection control varied by age and experience [26].

However, the lack of significant differences in attitudes by age, sex, nationality, or experience suggests that attitudinal barriers may be less pronounced than knowledge gaps. This is encouraging, as positive attitudes can facilitate the adoption of training interventions, but it also highlights the need to address organizational factors such as staffing and equipment availability, which can undermine confidence.

### 4.3. Practices in CRRT Management

The high percentage of low practice scores indicates limited proficiency in CRRT management, with a ceiling effect suggesting that the questionnaire may not fully capture the complexity of clinical practices. The weak positive correlation between knowledge and practices highlights the disconnect between understanding and application. This aligns with the findings of Alkubati et al. (2023), who noted that limited training and resources hinder CCNs’ ability to translate knowledge into practice [27]. Significant differences were observed in practice scores by age (>32 years), experience (>7 years), and patient-to-nurse ratio (1:1), indicating that workplace factors and individual characteristics influence performance. CCNs with a 1:1 ratio likely have more time to focus on CRRT tasks, thereby reducing errors and enhancing protocol adherence. Similarly, staff CCNs and CVVHDF users exhibited higher practice scores, reflecting their hands-on roles and familiarity with the prevalent modality. The lower practice scores among CCNs with extra outside work suggest that fatigue or divided attention may impair performance, a concern also raised by other studies regarding burnout among Saudi CCNs [28,29,30].

The high prevalence of CVVHDF use (65.3%) and the 1:1 nurse-to-patient ratio for CRRT (64.2%) indicate adherence to recommended standards, but the frequent tubing blockages reported by 11.1% of CCNs and their limited experience (83.2% with <3 years) highlight practical challenges. These findings echo those of Al Qahtani and Almetrek (2017) [25], who noted strong infection control practices but gaps in specific areas such as needle handling, suggesting that targeted training is needed to address CRRT-specific skills.

The results underscore significant training needs, particularly for CCNs with less experience, with lower educational qualifications, and in non-specialized departments (e.g., those in the ER, who had a mean score of 82.78 for knowledge). The moderate knowledge and limited practice scores align with the findings of Brost (2022), who demonstrated that a four-hour advanced CRRT training session significantly improved knowledge (*p* < 0.001) and confidence [31]. The weak knowledge–practice correlation suggests that current training may not adequately prepare CCNs for real-world CRRT challenges, such as managing alarms or adjusting parameters under pressure [7,12]. CCNs’ needs include hands-on training, simulation-based learning, and access to CRRT specialist panels, as recommended by Yu et al. (2024) [10]. Organizational support, such as standardized protocols and adequate staffing, is also critical, as high patient-to-nurse ratios and long shifts may exacerbate stress and errors [12,14]. The lack of specialist panels in some departments, inferred from practice assessment, further limits CCNs’ ability to seek expert guidance [32].

### 4.4. Contextual Factors in Saudi Arabia

The Saudi Arabian context, characterized by a diverse nursing workforce and varying training opportunities, influences the findings. The predominance of Saudi nationals (75.8%) and male CCNs (69.5%) reflects the local healthcare workforce; however, the lack of significant differences by nationality suggests that cultural diversity may not be a primary barrier to CRRT implementation. However, the high proportion of CCNs with bachelor’s degrees (81.6%) indicates a well-educated workforce, yet the lower knowledge scores among diploma and master’s holders suggest that CRRT-specific training is not adequately integrated at all educational levels. The reliance on 12-h shifts and extra outside work among 12.1% of CCNs highlights workload issues, which may contribute to burnout and reduced practice proficiency [28].

The increasing prevalence of AKI in Saudi Arabia [6] and the growing adoption of CRRT necessitate a robust nursing response. However, the limited research on CRRT in the kingdom, compared to that on hemodialysis (e.g., Alsolami and Alobaidi, 2024) [15], underscores the need for context-specific interventions. The findings suggest that Hail Hospitals, while equipped for CRRT, may lack the structured training programs and specialist support seen in teaching hospitals elsewhere, highlighting a gap in resource allocation [10].

#### 4.4.1. Implications for Practice and Policy

The results have several implications; for example, hospitals should implement regular hands-on CRRT training, focusing on practical skills such as alarm troubleshooting and vascular access management. Simulation-based learning, as demonstrated by Brost (2022), can enhance competency and confidence [31]. Developing and enforcing standardized CRRT protocols can reduce practice variations and ensure consistency across departments. This is particularly important given the high use of CVVHDF and the need for modality-specific guidelines. Optimizing patient-to-nurse ratios (e.g., maintaining a 1:1 ratio for CRRT) and establishing CRRT specialist panels can improve practice quality and reduce stress. Addressing long shifts and extra work through workload management policies is also critical. Continuous professional development through tailored programs for less experienced CCNs and those with diplomas can bridge knowledge gaps, while postgraduate curricula should incorporate CRRT training to prepare master’s-educated CCNs for clinical roles.

#### 4.4.2. Limitations

The strengths of this study include the use of a validated questionnaire, a robust sample size (N = 190), and a rigorous statistical analysis, providing reliable insights into CCNs’ KAP. However, limitations such as the use of convenience sampling can lead to sampling bias, which may limit the generalizability to other regions or hospitals. Another limitation is self-reporting bias, as CCNs may have over- or under-reported their KAP due to social desirability and recall biases. Although anonymity and confidentiality were assured, the potential overestimation of knowledge or practice proficiency cannot be excluded. Another limitation is the categorization of KAP levels into “low,” “moderate,” and “high”, which was based on percentile thresholds and prior validation but remained somewhat arbitrary. Using a cross-sectional study may capture a single point in time but limits insights into causal relationships or changes over time. Despite these limitations, this study provides valuable baseline evidence on CCNs’ readiness for CRRT in Saudi Arabia. Future research should employ longitudinal and interventional designs to evaluate the impact of structured CRRT training and use objective measures such as direct observation or simulation-based assessments to validate self-reported practices. Expanding the study to multiple regions and healthcare settings would also enhance generalizability.

## 5. Conclusions

This study revealed that CCNs in Hail Hospitals had high levels of knowledge and attitudes toward CRRT, followed by moderate levels, but exhibited limited practice proficiency. Knowledge and attitudes are influenced by experience, education, department, and CRRT modality, whereas practices are affected by age, workload, and job position. The moderate correlation between knowledge and attitude, contrasted with a weaker knowledge–practice link, indicates that, while CCNs understand CRRT principles, their practical application is hindered by training gaps and organizational constraints. Based on these findings, several clinical recommendations can be made. Hospitals should implement simulation-based and hands-on CRRT training programs to enhance practical competency; maintain safe nurse-to-patient ratios (e.g., 1:1 or 1:2 for patients receiving CRRT) to ensure effective monitoring; and develop modality-specific protocols, particularly for CVVHDF, to reduce variability and to standardize care. These findings highlight the urgent need for enhanced educational and institutional support to optimize CRRT implementation and improve patient outcomes.

## Figures and Tables

**Figure 1 healthcare-13-02320-f001:**
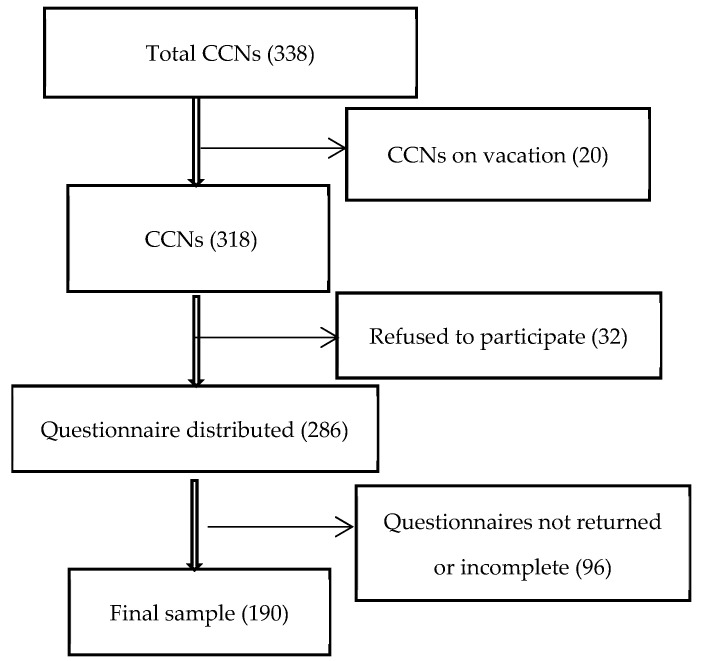
Flowchart of participant recruitment.

**Figure 2 healthcare-13-02320-f002:**
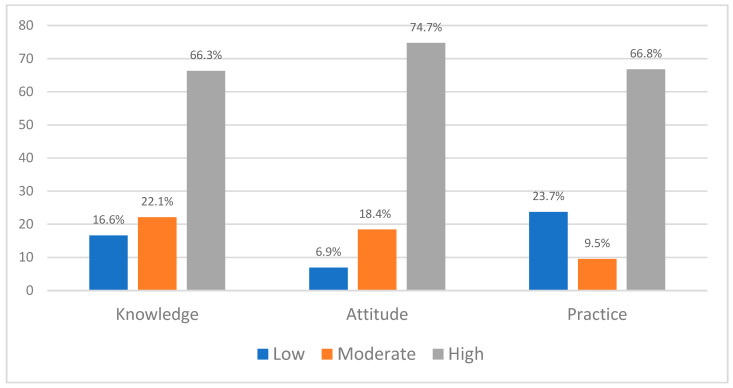
CCNs’ Knowledge, Attitudes, and Practices Toward CRRT.

**Table 1 healthcare-13-02320-t001:** Sociodemographic data of the participating CCNs (N = 190).

Variable		Count	(%)
Age (Year)	≤32	99	(52.1)
	>32	91	(47.9)
Sex	Male	132	(69.5)
	Female	58	(30.5)
Nationality	Saudi	144	(75.8)
	Non-Saudi	46	(24.2)
Experience (year)	≤7	107	(56.3)
	>7	83	(43.7)
Qualification	Diploma	17	(8.9)
	Bachelor	155	(81.6)
	Master	18	(9.5)
Hospital Department	Intensive Care Unit	67	(35.3)
	Emergency department (ER)	43	(22.6)
	Critical care unit (CCU)	38	(20.0)
	Artificial Kidney Unit (AKU)	42	(22.1)
Job position	Charge nurse	50	(26.3)
	Staff nurse	129	(67.9)
	Head nurse, supervisor	11	(5.8)
Patient-to-nurse ratio	1:1	28	14.8
	1:2	72	37.9
	1:3	39	20.5
	1:4	51	26.8
Shift hours	8 h	74	(38.9)
	12 h	116	(61.1)
Extra work outside	Yes	23	(12.1)
	No	167	(87.9)

**Table 2 healthcare-13-02320-t002:** CRRT Practice Patterns and CRRT Mode among CCNs (N = 190).

Variable		Count	(%)
Frequency of tubing blockage during CRRT	No attention	9	(4.7)
	Never seen before	84	(44.2)
	Sometimes	76	(40.0)
	Frequent	21	(11.1)
The CRRT mode you most frequently use	CVVH	20	(10.5)
	CVVHD	27	(14.2)
	CVVHDF	124	(65.3)
	SCUF	19	(10.0)

Abbreviations: CRRT, continuous renal replacement therapy; CVVH, continuous venovenous hemofiltration; CVVHD, continuous venovenous hemodialysis; CVVHDF, continuous venovenous hemodiafiltration; SCUF, slow continuous ultrafiltration.

**Table 3 healthcare-13-02320-t003:** Correlation between study variables (N = 190).

Variables		Knowledge	Attitudes	Practices
Knowledge	rs	1		
	*p*			
Attitudes	rs	0.230 **	1	
	*p*	<0.001		
Practices	rs	0.192 **	0.419 **	1
	*p*	<0.001	<0.001	

** Correlation is significant at the 0.01 level (2-tailed).

**Table 4 healthcare-13-02320-t004:** Differences in knowledge, attitude, and practice scores among CCNs (N = 190).

Variables		N	Knowledge		Attitude		Practice	
Age (Year)			Median IQR	*p* Value	Median IQR	*p* Value	Median IQR	*p* Value
Age (Year) ^T^	≤32	99	47 (38–50)	0.064	58 (52–61)	0.840	8 (4–9)	<0.001 *
	>32	91	48 (48–49)		58 (58–58)		10 (5–10)	
Sex ^T^	Male	132	48 (42–49)	0.725	58 (56–59)	0.270	9 (4–10)	0.276
	Female	58	48 (43–50)		58 (54–59)		8 (4–10)	
Nationality ^T^	Saudi	144	48 (41–49)	0.108	58 (56–59)	0.271	9 (9–10)	0.435
	Non-Saudi	46	49 (46–51)		58 (55–58)		8 (2–10)	
Experience (year) ^T^	≤7	107	47 (37–49)	<0.001 *	58 (48–61)	0.191	8 (8–10)	<0.001 *
	>7	83	49 (48–49)		58 (58–58)		10 (6–10)	
Shift hours	8 h	74	46 (38–49)	<0.001 *	58 (48–59)	0.437	8 (4–10)	0.035 *
	12 h	116	49 (48–50)		58 (57–58)		10 (5–10)	
Extra work outside ^T^	Yes	23	46 (38–51)	0.295	58 (52–60)	0.952	7 (7–9)	0.035 *
	No	167	48 (44–49)		58 (56–59)		9 (5–10)	
Qualification ^F^	Diploma	17	37 (24–59)	0.001 *	45 (29–73)	0.035 *	0 (0–0)	<0.001 *
	Bachelor	155	48 (47–49)		58 (57–59)		10 (7–10)	
	Master	18	39 (13–47)		51 (26–63)		5 (0–7)	
Hospital Department ^F^	ICU	67	49 (46–51)	<0.001 *	58 (56–60)	0.029 *	8 (6–10)	0.546
	ER	43	48 (41–49)		58 (53–59)		9 (2–10)	
	CCU	38	49 (48–51)		58 (58–65)		10 (0–10)	
	AKU	42	48 (34–48)		58 (48–58)		9 (2–10))	
Job position ^F^	Charge nurse	50	43 (28–52)	0.069	56 (45–61)	0.104	6 (0–8)	<0.001 *
	Staff nurse	129	48 (48–49)		58 (58–58)		10 (7–10)	
	Head nurse, supervisor	11	46 (26–49)		57 (57–59)		6 (3–9)	
Patient-to-nurse ratio ^F^	1:1	28	48 (46–49)	0.001 *	58 (53–58)	0.176	10 (7–10)	0.041 *
	1:2	72	49 (46–52)		58 (56–63)		8 (5–10)	
	1:3	39	48 (38–49)		58 (56–58)		10 (0–10)	
	1:4	51	47 (36–49)		58 (52–59)		9 (3–10)	
The CRRT mode you most frequently use ^F^	CVVH	20	38 (13–47)	<0.001 *	52 (15–58)	<0.001 *	4 (0–9)	<0.001 *
	CVVHD	27	42 (24–48)		54 (45–61)		8 (2–9)	
	CVVHDF	124	49 (48–49)		58 (58–59)		10 (7–10)	
	SCUF	19	38 (33–59)		48 (45–73)		0 (0–8)	

Note: Asterisks (*) indicate statistically significant differences. “T” indicates that the Mann–Whitney test was used; “F” indicates that the Kruskal–Wallis test was used. Abbreviations: CRRT, continuous renal replacement therapy; CVVH, continuous venovenous hemofiltration; CVVHD, continuous venovenous hemodialysis; CVVHDF, continuous venovenous hemodiafiltration; SCUF, slow continuous ultrafiltration; ICU, intensive care unit; CCU, critical care unit; ER, emergency department; Artificial Kidney Unit (AKU).

**Table 5 healthcare-13-02320-t005:** Multiple regression of factors affecting CCNs’ knowledge, attitude, and practice toward CRRT.

Factors		Knowledge			Attitudes			Practice		
		β	CI (95%)	*p*-Value	β	CI (95%)	*p*-Value	β	CI (95%)	*p*-Value
Age								−0.204	−0.461–0.053	0.119
Experience (year)		0.476	0.084–0.869	0.018				0.218	−0.088–0.523	0.162
Shift hours	8 h	Reference						Reference		
	12 h	−0.425	−4.794–3.943	0.848				0.694	−0.588–1.976	0.287
Extra work outside	Yes									
	No							0.369	−1.128–1.866	0.627
Qualification	Diploma	Reference			Reference			Reference		
	Bachelor	1.709	−4.382–7.800	0.581	7.916	1.596–14.236	0.014	4.639	2.432–6.846	<0.001
	Master	−4.350	−11.691–2.991	0.244	0.106	−7.421–7.633	0.978	2.385	−0.110–4.880	0.061
Hospital Department	ICU	Reference			Reference					
	ER	−0.723	−4.706–3.261	0.721	−0.151	−4.029–3.728	0.939			
	CCU	1.081	−2.957–5.118	0.598	3.358	−0.664–7.380	0.101			
	AKU	−1.489	−6.695–3.716	0.573	−0.418	−4.367–3.531	0.835			
Job position	Charge nurse									
	Staff nurse							1.143	−0.081–2.366	0.067
	Head nurse, supervisor							0.732	−1.617–3.081	0.539
Patient-to-nurse ratio	1:1	Reference						Reference		
	1:2	4.129	0.327–7.932	0.033				−0.984	−2.254–0.285	0.128
	1:3	2.264	−2.146–6.673	0.312				−0.573	−1.993–0.848	0.427
	1:4	2.871	−2.352–8.094	0.279				1.055	−0.723–2.833	0.243
The CRRT mode you most frequently use	CVVH	Reference			Reference			Reference		
	CVVHD	4.630	−1.084–10.345	0.112	7.599	1.705–13.492	0.012	2.004	0.099–3.909	0.039
	CVVHDF	11.699	6.334–17.065	<0.001	12.242	7.077–17.407	<0.001	2.704	0.838–4.569	0.005
	SCUF	7.325	0.394–14.257	0.038	12.191	5.182–19.201	0.001	0.969	−1.464–3.401	0.433

Abbreviations: β, unstandardized beta coefficient; CI; confidence interval, CRRT, continuous renal replacement therapy; CVVH, continuous venovenous hemofiltration; CVVHD, continuous venovenous hemodialysis; CVVHDF, continuous venovenous hemodiafiltration; SCUF, slow continuous ultrafiltration; ICU, intensive care unit; ER, emergency room; CCU, critical care unit; AKU, artificial kidney unit.

## Data Availability

The data presented in this study are available on request from the corresponding author.

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
