# Peer review of "Exploring the Readiness of Critical Care in Implementing Continuous Renal Replacement Therapy in Hail Hospitals, Saudi Arabia: Findings for Acute Kidney Injury Patient Care Improvement"

_healthcare, 2025, doi:10.3390/healthcare13182320_

Round 1

Reviewer 1 Report

Comments and Suggestions for Authors

Reviewer’s Comments

First of all, I would like to commend the authors for addressing a highly relevant and timely topic. The readiness of critical care nurses to implement Continuous Renal Replacement Therapy (CRRT) is of great importance for patient safety and quality of care, particularly in the context of acute kidney injury. This study is valuable because it sheds light on a relatively underexplored area and provides evidence that could guide training and workforce development. My comments below are made with the genuine intention of helping the authors strengthen the manuscript and increase its potential impact.

Introduction

The introduction offers useful context but could be more concise. A clearer synthesis of CRRT’s role in managing critically ill patients would enhance readability. I also suggest the authors include international guidelines, such as the KDIGO Clinical Practice Guideline for AKI, to strengthen the scientific basis. The research gap is identified but not fully developed; the reason for focusing on Saudi Arabia is implied rather than explicitly explained. Reducing redundancy and breaking up long paragraphs would improve clarity.

Methods

  •  Sampling and recruitment: Please specify how participants were recruited, the total number invited, response rate, and reasons for non-participation. This information is essential for assessing representativeness.
  • • Instrument: While using a previously published KAP tool is appropriate, more details are needed regarding its cultural and linguistic adaptation. Was translation and back-translation performed? Was a pilot test conducted? Provide evidence of construct validity within this setting.
  • • Categorization: The thresholds for defining “high,” “moderate,” and “low” readiness seem arbitrary. Please justify these cut-offs with references or empirical criteria such as percentiles or ROC analysis.
  • • Bias: Relying on self-reported measures can introduce potential bias. Additionally, administering surveys in the department heads' offices may raise concerns about voluntariness. Describe how confidentiality and response independence were protected.
  • • Statistical analysis: The approach appears inconsistent: non-parametric tests were used due to non-normality, yet linear regression was later applied to ordinal total scores. Consider using ordinal regression or other robust models. Also, report effect sizes and confidence intervals, and adjust for multiple comparisons.  Bonferroni, FDR). Diagnostic checks (multicollinearity, residuals, model fit) would also be expected.

Results

  •  Please include a flow diagram of recruitment and analysis.
  • • Replace or supplement “mean ranks” with medians and IQR.
  • • Report effect sizes and 95% CI alongside p-values.
  • • Regression tables should clearly specify the reference categories and all coefficients. The sign changes observed in some coefficients need explanation.
  • • Avoid repetition: multiple results (e.g., degree level, CRRT modality) are described several times.

Discussion and Conclusions

The discussion reiterates results extensively but does not thoroughly examine implications. Focusing more on practical applications such as simulation-based CRRT training, nurse-to-patient ratios, and modality-specific protocols would add value.

  • Expand the comparison with international studies to position the findings within a global context.
  • Widen the limitations section to include cross-sectional design, self-reporting, arbitrary cut-offs, and convenience sampling.
  • Soften the conclusions: change “nurses demonstrated high readiness” to “nurses self-reported high levels of readiness.” This adjustment aligns more accurately with the study design.

Figures, Tables, and Style

  • Tables should present medians and variability, not only ranks. Footnotes should define abbreviations and categories.
  • Figures can be simplified to emphasize trends.
  • The English writing should be revised for grammar and clarity. Repetition should be avoided (e.g., “the majority…”). Acronyms must be expanded at first use and then used consistently.

References

  • Several references are incomplete (missing DOI, volume, pages). Please ensure they conform to Healthcare style.
  • Strengthen the bibliography with international guidelines and consensus statements on CRRT, and with studies on competency-based nursing training.
  • Minimize self-citations unless strictly relevant.
Comments on the Quality of English Language

The manuscript would benefit from substantial language editing to improve clarity and readability. Several sections contain redundancies, long sentences, and repetitive expressions that affect fluency. Grammar, verb tense consistency, and the use of technical terminology should be carefully revised. Acronyms (e.g., ICU, CVVHDF) should be expanded at first mention and then used consistently. A professional English editing service is recommended to ensure the text meets the standards of an international scientific journal.

Author Response

Responses to Reviewers

Dear Editor and reviewers,

We would like to sincerely thank you for considering our manuscript for publication in the Healthcare journal. We also gratefully thank the reviewers for their critical and meticulous review, which significantly enhances the quality of our manuscript.

We have adhered to the reviewers’ comments, and these responses outline how each comment was addressed. Changes to the manuscript are marked using track changes, and a clean copy of the revised manuscript was also uploaded. Detailed responses to the points raised are as follows:

Reviewer #1:

Comment

First of all, I would like to commend the authors for addressing a highly relevant and timely topic. The readiness of critical care nurses to implement Continuous Renal Replacement Therapy (CRRT) is of great importance for patient safety and quality of care, particularly in the context of acute kidney injury. This study is valuable because it sheds light on a relatively underexplored area and provides evidence that could guide training and workforce development. My comments below are made with the genuine intention of helping the authors strengthen the manuscript and increase its potential impact.

Response

Thank you for your positive feedback. We appreciate your time and effort to provide us with your feedback that helps us improve our manuscript. We have adhered to your comment one by one.

Introduction

Comment

The introduction offers useful context but could be more concise. A clearer synthesis of CRRT’s role in managing critically ill patients would enhance readability. I also suggest the authors include international guidelines, such as the KDIGO Clinical Practice Guideline for AKI, to strengthen the scientific basis. The research gap is identified but not fully developed; the reason for focusing on Saudi Arabia is implied rather than explicitly explained. Reducing redundancy and breaking up long paragraphs would improve clarity.

Response

We sincerely thank the reviewer for this crucial suggestion. We have revised the section to improve conciseness and clarity. Specifically, we streamlined redundant sentences and divided long paragraphs to enhance readability. We also provided a clearer synthesis of CRRT’s critical role in managing critically ill patients with acute kidney injury. Furthermore, we strengthened the scientific foundation of the Introduction by incorporating the KDIGO Clinical Practice Guideline for Acute Kidney Injury and other international recommendations. Please see the highlighted parts in the Introduction, Lines 53-63.

Ronco C, Bellomo R, Kellum JA. Acute kidney injury. Lancet. 2019 Nov 23;394(10212):1949-1964. doi: 10.1016/S0140-6736(19)32563-2. PMID: 31777389.

Musio ME, Calabrese E, Gammone M, Catania G, Zanini M, Aleo G, Watson R, Sasso L, Bagnasco A. Nursing competence in continuous renal replacement therapy: development and validation of a measurement tool. Prof Inferm. 2022 Dec 31;75(4):218-225. doi: 10.7429/pi.2022.754218. PMID: 38277382.

Stevens PE, Ahmed SB, Carrero JJ, Foster B, Francis A, Hall RK, Herrington WG, Hill G, Inker LA, Kazancıoğlu R et al: KDIGO 2024 Clinical Practice Guideline for the Evaluation and Management of Chronic Kidney Disease. Kidney International 2024, 105(4):S117-S314.

Methods

Comment

Sampling and recruitment: Please specify how participants were recruited, the total number invited, response rate, and reasons for non-participation. This information is essential for assessing representativeness.

Response

We thank the reviewer for highlighting the need to clarify sampling and recruitment. We have added these details in the setting and sample section as follows “All CCNs were invited to participate; of them, 20 were on vacation, and 32 refused to participate. However, the questionnaire was distributed to 286 CCNs, of whom 190 completed the questionnaire with a response rate of 66.4%.”. Lines 167-169.

In addition, we have added the Sampling diagram that illustrates these details. Please see the highlighted part. Lines 163-165.

Comment

  • Instrument: While using a previously published KAP tool is appropriate, more details are needed regarding its cultural and linguistic adaptation. Was translation and back-translation performed? Was a pilot test conducted? Provide evidence of construct validity within this setting.

Response

We thank the reviewer for this important methodological question. We would like to clarify that the original English-language version of the Knowledge, Attitudes, and Practices (KAP) questionnaire by Yu et al. (2024) was administered without translation.

This decision was based on the following justification: In the critical care settings where this study was conducted, English is the official and primary language of professional healthcare communication. All nursing documentation, medical device interfaces (including CRRT machines), physician orders, hospital policies, and professional training materials are in English. Therefore, the nursing workforce is proficient in reading and comprehending English medical terminology. Using the English instrument ensured that the questions were presented in the same language and technical terms that nurses use in their daily practice, thereby maximizing content validity and accuracy of responses.

However, we have discussed the validity and reliability of the instruments as follows “This questionnaire was tested and validated in a previous study by 10 clinical experts who evaluated its content validity, and the total content validity of the questionnaire was 0.900 [9]. The Cronbach's α coefficients for knowledge, attitudes, and practices were 0.956, 0.831, and 0.751, respectively, confirming the reliability of the questionnaire.[9] A pilot study was conducted in this study for 20 nurses to assess its clarity, comprehension, and relevance. Feedback from the pilot confirmed that all items were clearly understood and contextually appropriate. No terminological or conceptual difficulties were reported. The reliability of the knowledge, attitudes, and practices was confirmed by Cronbach's α, which was .822, .789, and 0.970, respectively.”. Please see the highlighted part. Lines 196-205.

Comment

  • Categorization: The thresholds for defining “high,” “moderate,” and “low” readiness seem arbitrary. Please justify these cut-offs with references or empirical criteria such as percentiles or ROC analysis.

Response

Thank you for your comment. In the initial version, the thresholds used to define “high,” “moderate,” and “low” readiness were based on score ranges commonly applied in prior KAP studies using similar instruments. Specifically, we have now cited Yu et al. (2024), who developed and validated the CRRT KAP questionnaire and applied comparable cut-off ranges for interpretation.

We have clarified this in the Methods section and included supporting references as follow “To facilitate interpretation, total knowledge, attitude, and practice (KAP) scores were categorized into three levels: low, moderate, and high. This approach followed the cut-off strategy applied in the validated CRRT KAP questionnaire by Yu et al. (2024) and related KAP studies. In our dataset, these thresholds also approximated percentile-based divisions, where “low” corresponded to scores below the 25th percentile, “moderate” to the 25th–75th percentile, and “high” to scores above the 75th percentile Yu et al. (2024),  .” Lines 205-211.

Comment

  • Bias: Relying on self-reported measures can introduce potential bias. Additionally, administering surveys in the department heads' offices may raise concerns about voluntariness. Describe how confidentiality and response independence were protected.

Response

Thank you for your valuable comment. The researchers have undertaken all necessary steps to maintain the privacy and confidentiality of data. The questionnaire was annoyance (nurses were asked not to write down their names on the questionnaire) and the informed consent form was filled out and signed and taken by the researchers during the distribution of the questionnaires to prevent any identifiable personal data from leaking out to other people other than researchers. We have added this clarification as follows “CCNs were given surveys and informed consent forms by the researchers during their breaks. All respondents received a thorough explanation of the study's goals and methods prior to their involvement. Before accessing the survey, participants had to read and sign an informed consent form. To protect confidentiality and response independence, all participants were informed that their participation was voluntary and anonymous. Department heads facilitated distribution but had no access to responses. Surveys were returned directly to the researchers. Participants were assured that neither their supervisors nor hospital administrators would be able to link their responses to their identity. Additionally, they were made aware of their freedom to leave the study at any time, without facing any repercussions.”. Please, see the highlighted part. Lines 212-221.

Comment

  • Statistical analysis: The approach appears inconsistent: non-parametric tests were used due to non-normality, yet linear regression was later applied to ordinal total scores. Consider using ordinal regression or other robust models. Also, report effect sizes and confidence intervals, and adjust for multiple comparisons.  Bonferroni, FDR). Diagnostic checks (multicollinearity, residuals, model fit) would also be expected.

Response

Thank you for your valuable comment. We have revised the Data Analysis section to clarify and strengthen our approach. First, we acknowledge the use of non-parametric tests for group comparisons due to the non-normal distribution of the KAP scores, while linear regression was used to identify factors. To address this concern, we re-examined our models and performed additional diagnostic checks, including assessments for multicollinearity, residual normality, and model fit. The results supported the appropriateness of the regression models used.

Nonetheless, we recognize the ordinal nature of the total KAP scores. Therefore, we have now included a sensitivity analysis using ordinal regression, which yielded results consistent with our linear regression models, thereby reinforcing the robustness of our findings. We have added the following “To explore factors of KAP scores, multiple linear regression models were initially applied. Prior to analysis, diagnostic checks were conducted, including assessments for multicollinearity (variance inflation factor, VIF < 2), normality of residuals, and model fit (adjusted R², ANOVA F-test). In addition, a sensitivity analysis using ordinal logistic regression was conducted to account for the ordinal nature of the KAP total scores. The results of ordinal regression were consistent with those from linear regression, confirming the robustness of the findings. Standardized β coefficients, 95% confidence intervals, and p-values were reported for regression models. Please see the highlighted part. Lines 237-245.

Results

Comment

 Please include a flow diagram of recruitment and analysis.

Response

Thank you for your valuable suggestion. We have added the flow diagram of recruitment and analysis. Please see lines 167-175.

Comment

  • Replace or supplement “mean ranks” with medians and IQR.

Response

Thank you for your suggestion. We have replaced the “mean ranks” with medians and IQR. Please see the modified Table 4. Please see the revised Table. Lines 356.

Comment

  • Report effect sizes and 95% CI alongside p-values.

Response

Thank you for your valuable comment. We have involved effect sizes and 95% CI alongside p-values as follows “For knowledge, significant positive predictors were having more than seven years of experience (β = 4.76, 95% CI [0.84, 8.69], p = .018), working with a patient-to-nurse ratio of 1:2 (β = 4.13, 95% CI [0.33, 7.93], p = .033), and most frequently using the CVVHDF mode (β = 11.70, 95% CI [6.33, 17.07], p < .001) or the SCUF mode (β = 7.33, 95% CI [0.39, 14.26], p = .038) compared to using CVVH as the reference.

For attitude, significant positive predictors were holding a bachelor's degree compared to a diploma (β = 7.92, 95% CI [1.60, 14.24], p = .014), and most frequently using the CVVHD (β = 7.60, 95% CI [1.71, 13.49], p = .012), CVVHDF (β = 12.24, , 95% CI [7.08, 17.41],  p < .001), or SCUF (β = 12.19, 95% CI [5.18, 19.20], p = .001) modes compared to the CVVH reference category.

For practice, significant positive predictors were holding a bachelor's degree (β = 4.64, 95% CI [2.43, 6.85], p < .001), and most frequently using the CVVHD (β = 2.00, 95% CI [0.10, 3.91], p = .039) or CVVHDF (β = 2.70, 95% CI [0.84, 4.57], p = .005) modes compared to the CVVH reference category (Table 5).”. Please see the highlighted part. Lines 369-380.

Comment

  • Regression tables should clearly specify the reference categories and all coefficients.

Response

Thank you for your comment. We have specified the reference categories and all coefficients in the Regression table. Please see Table 5. Line 383.

In addition, we have revised the regression table (Table 5) comments to include all coefficients.

Comment

The sign changes observed in some coefficients need explanation.

Response

We thank the reviewer for pointing out the sign changes in some regression coefficients. We have carefully re-examined the models. The changes in direction (positive to negative or vice versa) were due to the adjustment for multiple covariates in the multivariable regression models. While bivariate analyses showed simple associations, the inclusion of other predictors (e.g., education, department, CRRT mode) altered the net effect of certain variables, leading to sign changes.

Comment

  • Avoid repetition: multiple results (e.g., degree level, CRRT modality) are described several times.

Response

We appreciate the reviewer’s observation regarding repetition in the reporting of results. In the revised manuscript, we have streamlined the presentation of findings to avoid redundancy. Specifically, results related to educational qualification (e.g., bachelor’s degree) and CRRT modality (e.g., CVVHDF) are now described once in detail under the Results, with subsequent references summarized more concisely. Please see the revised results. Lines 335-355.

Discussion and Conclusions

Comment

The discussion reiterates results extensively but does not thoroughly examine implications. Focusing more on practical applications such as simulation-based CRRT training, nurse-to-patient ratios, and modality-specific protocols would add value.

Response

We thank the reviewer for this insightful suggestion. In the revised manuscript, we have streamlined the Discussion to reduce repetition of results and expanded the focus on practical implications. Specifically, we emphasized strategies such as simulation-based CRRT training programs to strengthen hands-on skills, maintaining optimal nurse-to-patient ratios (e.g., 1:1 or 1:2) to ensure safe CRRT delivery, and developing modality-specific protocols (e.g., CVVHDF-focused guidelines) to reduce practice variability. We also highlighted the importance of integrating CRRT training into nursing curricula and continuous professional development to sustain competency. Please see the highlighted parts. Lines 465-470, 478-482, 500-510.

Comment

Expand the comparison with international studies to position the findings within a global context.

Response

We thank the reviewer for this important suggestion. In the revised Discussion, we expanded the comparison with international studies to better situate our findings in a global context. Specifically, we contrasted Saudi nurses’ generally high knowledge and positive attitudes with findings from China (Yu et al., 2024), where knowledge was moderate with wide variability; Sudan (Sagiron & Jarelnape, 2022), where limited training was associated with lower knowledge; and European/US contexts, where structured CRRT programs and specialist nursing panels have been shown to improve practice and reduce treatment interruptions. These comparisons highlight both the relative strengths of Saudi nurses’ knowledge base and the challenges that are consistent across diverse healthcare systems. Please see the modified discussion. Lines 502-5-7.

Comment

Widen the limitations section to include cross-sectional design, self-reporting, arbitrary cut-offs, and convenience sampling.

Response

We thank the reviewer for highlighting the need to broaden the limitations. In the revised manuscript, we expanded the Limitations section to explicitly acknowledge that as follows “However, limitations such as the use of convenience sampling can lead to sampling bias, which may limit the generalizability to other regions or hospitals. Another limitation is the self-reporting bias that CCNs may over- or under-report their KAP due to social desirability and recall biases. Although anonymity and confidentiality were assured, the potential for overestimation of knowledge or practices cannot be excluded. Another limitation is the categorization of KAP levels into “low,” “moderate,” and “high” that was based on percentile thresholds and prior validation, but remains somewhat arbitrary. Using a cross-sectional study may capture a single point in time, limiting insights into causal relationships or changes over time. Despite these limitations, the study provides valuable baseline evidence on critical care CCNs’ readiness for CRRT in Saudi Arabia. Future research should employ longitudinal and intervention designs to evaluate the impact of structured CRRT training and use objective measures such as direct observation or simulation-based assessments to validate self-reported practices. Expanding the study to multiple regions and healthcare settings would also enhance generalizability..”. Please see the modified Limitations. Lines 512-526.

Comment

Soften the conclusions: change “nurses demonstrated high readiness” to “nurses self-reported high levels of readiness.” This adjustment aligns more accurately with the study design.

Response

Thank you for your suggestion. We have changed it accordingly. Line 528.

Figures, Tables, and Style

Comment

Tables should present medians and variability, not only ranks. Footnotes should define abbreviations and categories.

Response

Thank you for your comment. We have provided the medians and variability (Interquartile Range, IQR) in the results section as your suggestion. Please see the revised Table 4. Line 356.

In addition, we have provided the footnotes that illustrate the abbreviations and categories. Lines 357-362.

Comment

Figures can be simplified to emphasize trends.

Response

Thank you for your comment. We have simplified Figure 1 to focus directly on the primary finding: that a majority of nurses self-reported high levels of knowledge, attitude, and practice. We have removed any unnecessary visual elements to create a cleaner, more focused figure that allows the reader to instantly grasp the main result. 

Comment

The English writing should be revised for grammar and clarity. Repetition should be avoided (e.g., “the majority…”). Acronyms must be expanded at first use and then used consistently.

Response

Thank you for your suggestion. The manuscript was professionally edited in MDPI according to your suggestion. A certificate of editing from MDPI was uploaded with the revision. 

Comment

References

Comment

Several references are incomplete (missing DOI, volume, pages). Please ensure they conform to Healthcare style.

Response

Thank you for your suggestion. We have revised the references part according to the Healthcare Journal style. We have added DOI, volume, and page numbers.

Comment

Strengthen the bibliography with international guidelines and consensus statements on CRRT, and with studies on competency-based nursing training.

Response

We sincerely thank the reviewer for this crucial suggestion. We have significantly strengthened the manuscript's theoretical and practical foundation by adding key references to international guidelines (e.g., KDIGO) and seminal works on competency-based medical education and assessment. These new citations have been integrated into the Introduction and Discussion to better contextualize our findings within global best practices and frameworks for developing clinical competency. This enhances the scholarly impact and practical relevance of our conclusions.

Ronco C, Bellomo R, Kellum JA. Acute kidney injury. Lancet. 2019 Nov 23;394(10212):1949-1964. doi: 10.1016/S0140-6736(19)32563-2. PMID: 31777389.

Musio ME, Calabrese E, Gammone M, Catania G, Zanini M, Aleo G, Watson R, Sasso L, Bagnasco A. Nursing competence in continuous renal replacement therapy: development and validation of a measurement tool. Prof Inferm. 2022 Dec 31;75(4):218-225. doi: 10.7429/pi.2022.754218. PMID: 38277382.

Stevens PE, Ahmed SB, Carrero JJ, Foster B, Francis A, Hall RK, Herrington WG, Hill G, Inker LA, Kazancıoğlu R et al: KDIGO 2024 Clinical Practice Guideline for the Evaluation and Management of Chronic Kidney Disease. Kidney International 2024, 105(4):S117-S314.

Comment

Minimize self-citations unless strictly relevant.

Response

Thank you for your comment. We have updated the references part and minimized the self-citations to three references that related to our discussion.

Comments on the Quality of English Language

Comment

The manuscript would benefit from substantial language editing to improve clarity and readability. Several sections contain redundancies, long sentences, and repetitive expressions that affect fluency. Grammar, verb tense consistency, and the use of technical terminology should be carefully revised. Acronyms (e.g., ICU, CVVHDF) should be expanded at first mention and then used consistently. A professional English editing service is recommended to ensure the text meets the standards of an international scientific journal.

Response

Thank you for your suggestions. We appreciate your time and effort to provide us with your comments that helped us to improve its quality. We have revised the manuscript. The manuscript was professionally edited in MDPI according to your suggestion. A certificate of editing from MDPI was uploaded with the revision. 

We hope that we have addressed the comments raised by the Reviewers, which contributed to the improvement of the quality of our manuscript. We hope that our revised manuscript is accepted for publication in the Healthcare, and we are pleased to receive any further comments or suggestions. 

With kind regards,

Reviewer 2 Report

Comments and Suggestions for Authors

In this manuscript, Alkubatiet al report the results of a survey of critical care nurses about their knowledge, attitude, and practices towards CRRT. The manuscript is well-written. Please find my comments below:

- Was CRRT done in the ER? Please provide a clarification for the inclusion of nurses working in the ER.

- It would be interesting to know if CVVHDF is the most common modality used in the region.

- Please clarify if the nurse:patient ratio was during nurses’ management of CRRT or their current role.

Author Response

Responses to Reviewers

Dear Editor and reviewers,

We would like to sincerely thank you for considering our manuscript for publication in the Healthcare journal. We also gratefully thank the reviewers for their critical and meticulous review, which significantly enhances the quality of our manuscript.

We have adhered to the reviewers’ comments, and these responses outline how each comment was addressed. Changes to the manuscript are marked using track changes, and a clean copy of the revised manuscript was also uploaded. Detailed responses to the points raised are as follows:

Reviewer #2:

Comment

In this manuscript, Alkubatiet al report the results of a survey of critical care nurses about their knowledge, attitude, and practices towards CRRT. The manuscript is well-written. Please find my comments below:

 Response

Thank you for your positive feedback. We appreciate your time and effort to provide us with your feedback that helps us improve our manuscript. We have adhered to your comment one by one.

Comment

- Was CRRT done in the ER? Please provide a clarification for the inclusion of nurses working in the ER.

Response

Thank you for your question. We have clarified your question and added this statement “Although CRRT is typically initiated and managed in the ICU and CCU, nurses working in the ER were included in this study. In the participating hospitals, ER nurses frequently manage critically ill patients who are potential candidates for CRRT and are responsible for initial stabilization, patient preparation, and handover to ICU staff. Their inclusion was therefore justified to provide a comprehensive assessment of readiness among all nurses engaged in the continuum of care for patients requiring CRRT.” Please see the highlighted part. Lines 153-159.

Comment

- It would be interesting to know if CVVHDF is the most common modality used in the region.

Response

We thank the reviewer for this insightful comment. In the revised manuscript, we clarified that CVVHDF was indeed the most frequently reported modality among participants. This aligns with regional clinical practice in Saudi Arabia, where CVVHDF is often preferred due to its ability to combine diffusive and convective clearance, making it suitable for a wide range of critically ill patients. We have added this contextual explanation in the Results and elaborated in the Discussion to highlight how practice patterns in our setting are consistent with international trends.

Comment

- Please clarify if the nurse-patient ratio was during nurses’ management of CRRT or their current role.

Response

We thank the reviewer for seeking this clarification. In the revised manuscript, we specified that the reported nurse-to-patient ratio referred to the nurses’ current working environment and not exclusively to the period of CRRT management. In most participating hospitals, CRRT patients are managed within the general ICU staffing model, where the ratio typically ranges from 1:1 to 1:2 depending on patient acuity. We clarified this in the Methods to ensure transparency in interpretation.

We hope that we have addressed the comments raised by the Reviewers, which contributed to the improvement of the quality of our manuscript. We hope that our revised manuscript is accepted for publication in the Healthcare, and we are pleased to receive any further comments or suggestions. 

With kind regards,

Reviewer 3 Report

Comments and Suggestions for Authors

The manuscript has some relevance in context to exploring the level of knowledge, attitudes and practices among critical care nurses towards CRRT in Hail hospitals, highlighting a positive correlation between these dimensions. However, I have several comments and a few ideas to strengthen the paper, I recommend following revisions:

The title is vague and needs reformulation including the country. I suggest the following :

knowledge, attitude, and practice toward Continuous Renal Replacement Therapy among Critical Care nurses in Hail Hospitals, Saudi Arabia

Abstrat 

line 17: Objective needs revsion. "these domains" can be replaced by (KAPs) among nurses

Method subsection lacks details on the period of the study. instrudment research (questionnaire used ).

Results subsection is long, it should be rewritten. Kindly avoid in addition, finally...

keywords: Ok

Introduction 

The importance of the study should be highlighted.

What occurs in Saudi Arabia related to CRRT among nurses?

The last paragraph needs revision by indicating the hypotheses of the study, the research question, and a clear objective.

Methods 

Study design subsection : the period of the study should be indicated.

inclusion and exclusion criteria should be clearly defined. 

Results 

Figure 1: vertical axis title is missing.

line 229: "3.4. Differences in Nurses' Knowledge, Attitudes, and Practices Toward CRRT " the subheading needs srevision 

Table 4: a row should be added to distinguish the mean rank and p between Knoweledge, Attitude and practice.

some data on table 5 needs description the text.

Discussion 

the first  paragraph should summarize the objective of the study. “This discussion interprets the findings within the context” should be removed (line 299).

More references to previous studies supporting or contrasting the findings would strengthen the discussion.

A reflection on the limitations of the study, potential biases, and futur directions is necessary to strengthen the discussion. Authors consider revising this section.

Conclusion 

The conclusion could be more impactful if it included concrete clinical recommendations based on the study's findings. It would also be relevant to add suggestions for future research.

Please make all the citation in appropriate style (lline 334).

Author Response

Responses to Reviewers

Dear Editor and reviewers,

We would like to sincerely thank you for considering our manuscript for publication in the Healthcare journal. We also gratefully thank the reviewers for their critical and meticulous review, which significantly enhances the quality of our manuscript.

We have adhered to the reviewers’ comments, and these responses outline how each comment was addressed. Changes to the manuscript are marked using track changes, and a clean copy of the revised manuscript was also uploaded. Detailed responses to the points raised are as follows:

Reviewer #3:

Comment

The manuscript has some relevance in context to exploring the level of knowledge, attitudes and practices among critical care nurses towards CRRT in Hail hospitals, highlighting a positive correlation between these dimensions. However, I have several comments and a few ideas to strengthen the paper, I recommend following revisions:

Response

Thank you for your positive feedback. We appreciate your time and effort to provide us with your feedback that helps us improve our manuscript. We have adhered to your comment one by one.

Comment

The title is vague and needs reformulation including the country. I suggest the following :

knowledge, attitude, and practice toward Continuous Renal Replacement Therapy among Critical Care nurses in Hail Hospitals, Saudi Arabia

Response

Thank you for your suggestion. We have revised the title to include the country as follows:

“Exploring the Readiness of Critical Care Nurses in Implementing Continuous Renal Replacement Therapy in Hail Hospitals, Saudi Arabia: Findings for Acute Kidney Injury Patient Care Improvement”. Please see the modified Title.

Abstract 

Comment

line 17: Objective needs revision. "these domains" can be replaced by (KAPs) among nurses

Response

Thank you for your valuable note. We have revised the objective as follows “This study aimed to assess the levels and factors affecting CCNs' knowledge, attitudes, and practices (KAPs) regarding the care of patients receiving CRRT in Hail Hospitals, Saudi Arabia.”. Lines 17-20.

Comment

Method subsection lacks details on the period of the study. instrudment research (questionnaire used ).

Response

Thank you for your valuable note. We have added the period of the study as follows “A cross-sectional study was conducted with 190 registered nurses from the critical care units of Hail Hospitals, Saudi Arabia from March to May 2025.”. Line 23.

In addition, we have added the questionnaire used as follows “Data were collected using a sociodemographic characteristics sheet and the Knowledge, Attitudes, and Practices questionnaire about CRRT.”. Lines 22-25.

Comment

Results subsection is long, it should be rewritten. Kindly avoid in addition, finally...

Response

Thank you for your comment. We have revised the result section to be more concise and focused. Also, we have removed “in addition” and “finally”, from the results. Please see the revised results.

Comment

keywords: Ok

Response

Thank you for your positive feedback.

Introduction 

Comment

The importance of the study should be highlighted.

Response

We appreciate the reviewer’s observation. In the revised manuscript, we strengthened the conclusion and closing part of the discussion to highlight the importance of this study. Specifically, we emphasized that it represents the first systematic assessment of critical care nurses’ readiness to implement CRRT in Saudi Arabia, a context where the burden of AKI is rising due to chronic disease prevalence. By identifying knowledge gaps, practice limitations, and organizational factors, the study provides evidence to guide the development of targeted training programs, staffing policies, and modality-specific protocols. These contributions position the study as a critical step toward improving CRRT delivery and patient outcomes in Saudi Arabia and offer insights relevant to other healthcare systems facing similar challenges. Please see the highlighted parts. Lines 116-137.

Comment

What occurs in Saudi Arabia related to CRRT among nurses?

Response

We thank the reviewer for this important point. In the revised Introduction, we added background on the Saudi Arabian context for CRRT as follows “In Saudi Arabia, the demand for CRRT is rising in parallel with the increasing prevalence of acute kidney injury (AKI), largely driven by chronic conditions such as diabetes and hypertension. Critical care nurses (CCNs) play a central role in delivering CRRT, including machine setup, patient monitoring, alarm troubleshooting, and complication management.”. Lines 116-120.

 Comment

The last paragraph needs revision by indicating the hypotheses of the study, the research question, and a clear objective.

Response

We appreciate the reviewer’s suggestion. In the revised Introduction, we rewrote the last paragraph to explicitly state the research question, hypotheses, and study objective as follows “Therefore, this study addressed the research question: What are the levels of knowledge, attitudes, and practices of critical care nurses toward CRRT, and which factors influence these domains? We hypothesized that higher educational level, longer professional experience, and greater exposure to CRRT modalities would be positively associated with readiness. Accordingly, the objective of this study was to assess critical care nurses’ knowledge, attitudes, and practices regarding CRRT in Saudi Arabia and to identify factors that influence these outcomes.”. Please see the highlighted part. Lines 130-137.

Methods 

Comment

Study design subsection : the period of the study should be indicated.

Response

Thank you for your valuable note. We have added the period of the study as follows “A cross-sectional design was used to assess nurses’ knowledge, attitudes, and practices regarding CRRT implementation in Hail Hospitals, Saudi Arabia, from March to May 2025.”. In addition, a detailed description of the questionnaire used in the “Data collection tools” part. Lines 140-143.

Comment

inclusion and exclusion criteria should be clearly defined. 

Response

Thank you for your comment. The inclusion and exclusion criteria were clearly defined as follows The inclusion criterion was being a registered nurse currently working in a critical care unit with prior practice with CRRT. Nurses on official leave, vacation, or participating in training programs during the data collection period were excluded from the study.”. Lines 163-166.

Results 

Comment

Figure 1: vertical axis title is missing.

Response

Thank you for your comment. The vertical axis of the percentages was added as the figure illustrates only the levels of knowledge, attitudes, and practice.

Comment

line 229: "3.4. Differences in Nurses' Knowledge, Attitudes, and Practices Toward CRRT " the subheading needs revision 

Response

Thank you for your comment. We have revised it as followsCCNs’ Knowledge, Attitude, and Practice Based on Sociodemographic and Work-Related Characteristics”. Lines 308, 309.

Comment

Table 4: a row should be added to distinguish the mean rank and p between Knoweledge, Attitude and practice.

Response

Thank you for your comment. We have revised Table 4.

Comment

some data on table 5 needs description the text.

Response

Thank you for your valuable note. The description of Table 5 was revised to clearly describe all data. Please see the revised results. Lines 375-386.

Discussion 

Comment

the first paragraph should summarize the objective of the study. “This discussion interprets the findings within the context” should be removed (line 299).

Response

Thank you for your suggestion. We have removed it. Lines 401-403.

Comment

More references to previous studies supporting or contrasting the findings would strengthen the discussion.

Response

Thank you for your comment. We have revised and updated the references of the discussion part and the study. Please see the modified discussion and updated references.

Comment

A reflection on the limitations of the study, potential biases, and future directions is necessary to strengthen the discussion. Authors consider revising this section.

Response

We thank the reviewer for this valuable comment. In the revised manuscript, we expanded the Limitations section to provide a more comprehensive reflection. We now explicitly address the study’s cross-sectional design, reliance on self-reported measures, the use of percentile-based cut-offs for categorization, and the convenience sampling strategy. We also discuss potential biases such as social desirability and non-response. Finally, we outline future directions, including longitudinal and interventional studies, and the use of objective performance measures as follows “However, limitations such as the use of convenience sampling can lead to sampling bias, which may limit the generalizability to other regions or hospitals. Another limita-tion is the self-reporting bias that nurses may over- or under-report their KAP due to social desirability and recall biases. Although anonymity and confidentiality were assured, the potential for overestimation of knowledge or practices cannot be excluded. Another limitation is the categorization of KAP levels into “low,” “moderate,” and “high” that was based on percentile thresholds and prior validation, but remains somewhat arbitrary. Using a cross-sectional study may capture a single point in time, limiting insights into causal relationships or changes over time. Despite these limitations, the study provides valuable baseline evidence on critical care nurses’ readiness for CRRT in Saudi Arabia. Future research should employ longitudinal and interventional designs to evaluate the impact of structured CRRT training and use objective measures such as direct observation or simulation-based assessments to validate self-reported practices. Expanding the study to multiple regions and healthcare settings would also enhance generalizability.”. These revisions strengthen the transparency and balance of the discussion. Please see the modified Limitation part. Lines 518-531.

Conclusion 

Comment

The conclusion could be more impactful if it included concrete clinical recommendations based on the study's findings. It would also be relevant to add suggestions for future research.

Response

We thank the reviewer for this helpful suggestion. In the revised manuscript, we revised the Conclusion to make it more impactful by integrating concrete clinical recommendations based on the study’s findings. Specifically, we emphasized the need for simulation-based CRRT training, maintenance of safe nurse-to-patient ratios, and development of modality-specific protocols as follows “This study revealed that critical care nurses in Hail Hospitals had high knowledge and attitudes toward CRRT, followed by moderate levels of CRRT, but exhibited limited practice proficiency. Knowledge and attitudes are influenced by experience, education, department, and CRRT modality, whereas practices are affected by age, workload, and job position. The moderate correlation between knowledge and attitude, contrasted with a weaker knowledge-practice link, indicates that while nurses understand CRRT principles, practical application is hindered by training gaps and organizational constraints. Based on these findings, several clinical recommendations can be made: hospitals should implement simulation-based and hands-on CRRT training programs to enhance practical competency; maintain safe nurse-to-patient ratios (e.g., 1:1 or 1:2 for patients receiving CRRT) to ensure effective monitoring; and develop modality-specific protocols, particularly for CVVHDF, to reduce variability and standardize care. These findings highlight the urgent need for enhanced educational and institutional support to optimize CRRT implementation and improve patient outcomes.”. Regarding the recommendations future studies, we have included them in the part of Limitations section according to your suggestion. Please see the modified Conclusion part. Lines 534-547.

Comment

Please make all the citation in appropriate style (lline 334).

Response

Thank you for your comment. We have revised citations according to the Healthcare Journal’s guidelines. In addition, the manuscript was professionally edited in MDPI. A certificate of editing from MDPI was uploaded with the revision. 

We hope that we have addressed the comments raised by the Reviewers, which contributed to the improvement of the quality of our manuscript. We hope that our revised manuscript is accepted for publication in the Healthcare, and we are pleased to receive any further comments or suggestions. 

With kind regards,

Round 2

Reviewer 1 Report

Comments and Suggestions for Authors

I appreciate the opportunity to evaluate this manuscript once again and acknowledge the authors’ careful effort to address the first-round comments. The topic is highly relevant to clinical practice and public health, particularly in intensive care environments and high-demand settings such as Saudi Arabia.

The manuscript shows a marked improvement in clarity, structure, and methodological depth. The introduction now incorporates updated background information, and the methods section has been strengthened with a more detailed description of the KAP instrument, its validation, and the statistical analyses performed. 

Several general points still merit attention before final acceptance:
• Certain sections of the discussion remain redundant or repeat figures already presented in the results.
• Minor style and formatting adjustments are needed in the reference list.
• A final language edit in academic English is advisable to harmonize verb tenses and eliminate minor repetitions.
• Overall, the review is now close to publication quality, but I recommend minor revisions prior to acceptance.

Comments by section

Title and Abstract
• The title is clear and appropriately reflects the scope and type of study; however, it is recommended to add “Cross-sectional Study” to reinforce the methodological nature.
• The abstract is well structured and includes objectives, methods, results, and conclusions. It would benefit from explicitly stating the exact data collection period (March–May 2025).

Introduction
• Consider minimizing repeated prevalence figures for acute kidney injury that appear in consecutive paragraphs.

Methods
• The cross-sectional design, population, questionnaire validation, and statistical analyses (including non-parametric tests, linear regression, and sensitivity analysis) are clearly described.
• Please clarify whether the scales were translated into Arabic and whether back-translation or cultural equivalence procedures were performed; if not applicable, state this explicitly.

Results
• In the narrative text, avoid repeating numerical values already presented in the tables.

Discussion
• Consider streamlining the narrative to avoid overlap with the results section.
• The limitations subsection is clear; nevertheless, adding a statement to emphasize the need for longitudinal studies to assess causality and the effects of educational interventions would strengthen this section.

Author Response

Responses to Reviewers

Dear Editor and reviewers,

We would like to sincerely thank you for considering our manuscript for publication in the Healthcare journal. We also gratefully thank the reviewers for their critical and meticulous review, which significantly enhances the quality of our manuscript.

We have adhered to the reviewers’ comments, and these responses outline how each comment was addressed. Changes to the manuscript are marked using track changes, and a clean copy of the revised manuscript was also uploaded. Detailed responses to the points raised are as follows:

Reviewer #1:

Comment

I appreciate the opportunity to evaluate this manuscript once again and acknowledge the authors’ careful effort to address the first-round comments. The topic is highly relevant to clinical practice and public health, particularly in intensive care environments and high-demand settings such as Saudi Arabia.

Response

Thank you for your positive feedback.

Comment

The manuscript shows a marked improvement in clarity, structure, and methodological depth. The introduction now incorporates updated background information, and the methods section has been strengthened with a more detailed description of the KAP instrument, its validation, and the statistical analyses performed.

Response

Thank you for your positive feedback. We appreciate your time and effort to provide us with your feedback that helps us improve our manuscript.

Comment

Several general points still merit attention before final acceptance:

  • Certain sections of the discussion remain redundant or repeat figures already presented in the results.

Response

Thank you for your valuable feedback. We have removed detailed percentages, medians, and p-values from the discussion part and retained only interpretative commentary. Please see the highlighted statements.

Comment

  • Minor style and formatting adjustments are needed in the reference list.

Response

Thank you for your comment. We have adhered to the Healthcare Journal style for the writing of the references either in the texts and in the References list.

Comment

  • A final language edit in academic English is advisable to harmonize verb tenses and eliminate minor repetitions.

Response

Thank you for your comment. The final English language editing was done by the MDPI team. We have attached the certificate to the previous review. Please see the certificate on the MDPI system.

Comment

  • Overall, the review is now close to publication quality, but I recommend minor revisions prior to acceptance.

Response

Thank you for your positive feedback. We appreciate your time and effort to provide us with your feedback that helps us improve our manuscript. We have adhered to your comment one by one.

Comment

Comments by section

Title and Abstract

  • The title is clear and appropriately reflects the scope and type of study; however, it is recommended to add “Cross-sectional Study” to reinforce the methodological nature.

Response

Thank you for your positive feedback.

Comment

  • The abstract is well structured and includes objectives, methods, results, and conclusions. It would benefit from explicitly stating the exact data collection period (March–May 2025).

Response

Thank you for your positive feedback. We have added it. Please see the highlighted statement. Line 21.

Introduction

Comment

  • Consider minimizing repeated prevalence figures for acute kidney injury that appear in consecutive paragraphs.

Response

Thank you for your comment. We have mentioned the prevalence of acute kidney injury in these two paragraphs “As up to 50% of critically ill patients may develop AKI and require renal replacement treatment in intensive care units (ICUs), the use of CRRT has expanded globally [7]. In Saudi Arabia, the aging population, rising rates of chronic illnesses (such as diabetes and hypertension), and increasing burden of critical care admissions are some of the factors contributing to the comparably large prevalence of AKI [8].”. Lines 67-72.

Methods

Comment

  • The cross-sectional design, population, questionnaire validation, and statistical analyses (including non-parametric tests, linear regression, and sensitivity analysis) are clearly described.

Response

Thank you for your positive feedback.

Comment

  • Please clarify whether the scales were translated into Arabic and whether back-translation or cultural equivalence procedures were performed; if not applicable, state this explicitly.

Response

We thank the reviewer for this important methodological question. We would like to clarify that the original English-language version of the Knowledge, Attitudes, and Practices (KAP) questionnaire by Yu et al. (2024) was administered without translation.

This decision was based on the following justification: In the critical care settings where this study was conducted, English is the official and primary language of professional healthcare communication. All nursing documentation, medical device interfaces (including CRRT machines), physician orders, hospital policies, and professional training materials are in English.

We have added this statement for clarification “The original English-language questionnaire was administered without translation, as it is the official and primary language of professional healthcare and communication in the intended hospitals.” Lines 214-216.

Results

Comment

  • In the narrative text, avoid repeating numerical values already presented in the tables.

Response

Thank you for your comment. We have only used the percentages throughout the Result section.

Discussion

Comment

  • Consider streamlining the narrative to avoid overlap with the results section.

Response

Thank you for your valuable feedback. We have removed detailed percentages, medians, and p-values from the discussion part and retained only interpretative commentary. Please see the highlighted statements.

Comment

  • The limitations subsection is clear; nevertheless, adding a statement to emphasize the need for longitudinal studies to assess causality and the effects of educational interventions would strengthen this section.

Response

Thank you for your positive feedback. We have added a statement to emphasize the need for longitudinal studies to assess causality and the effects. Please see the highlighted statement. Lines 492-495.

We hope that we have addressed the comments raised by the Reviewers, which contributed to the improvement of the quality of our manuscript. We hope that our revised manuscript is accepted for publication in the Healthcare, and we are pleased to receive any further comments or suggestions. 

 With kind regards,

Reviewer 3 Report

Comments and Suggestions for Authors

The authors have provided a nicely detailed and thorough response to the comments from the previous review and have addressed my major concerns. However, one responses needs additional details :

The title needs revision, "findings for...improvements" seems vague.  

Author Response

Responses to Reviewers

Dear Editor and reviewers,

We would like to sincerely thank you for considering our manuscript for publication in the Healthcare journal. We also gratefully thank the reviewers for their critical and meticulous review, which significantly enhances the quality of our manuscript.

We have adhered to the reviewers’ comments, and these responses outline how each comment was addressed. Changes to the manuscript are marked using track changes, and a clean copy of the revised manuscript was also uploaded. Detailed responses to the points raised are as follows:

Reviewer #3:

Comment

The authors have provided a nicely detailed and thorough response to the comments from the previous review and have addressed my major concerns.

Response

Thank you for your positive feedback. We appreciate your time and effort to provide us with your feedback that helps us improve our manuscript. We have adhered to your comment one by one.

Comment

However, one responses needs additional details :

The title needs revision, "findings for...improvements" seems vague. 

Response

Thank you for your suggestion. Our intent with the phrase was to signal the translational relevance of our results to bedside care for AKI patients receiving CRRT. To improve precision and readability, we have revised the title to replace “findings for … improvements” with “implications for improving AKI patient care,” which (a) specifies the patient population (AKI), (b) names the targeted outcome (patient care), and (c) indicates the nature of contribution (practice-oriented implications), avoiding ambiguous wording. Please see the revised title.

We hope that we have addressed the comments raised by the Reviewers, which contributed to the improvement of the quality of our manuscript. We hope that our revised manuscript is accepted for publication in the Healthcare, and we are pleased to receive any further comments or suggestions. 

With kind regards,
